



# Coupling physics-informed neuronal networks with 3D scanning pulsed Doppler lidar

Christian Schiefer[1], Sebastian Kauczok[1], Albert Töws[1], Andre Weipert[1]

Leonardo Germany GmbH[1]

*Correspondence to*: c.schiefer@leonardogermany.com

**Abstract.** Physics-Informed neuronal networks (PINN) is a research field where a neuronal network is trained to solve an incorporated partial differential equation that describes some physical phenomenon. This work describes the coupling of the Navier Stokes (NS) equation with data from a 3D scanning pulsed Doppler lidar to reconstruct blanked sectors with radial velocities in a plan position indicator (PPI) scan. For the reconstruction, only the adjacent line of sight (LOS) measurements

were used as input data for the neuronal network. Almost one year of collected lidar data were used to analyze the wind field sector reconstruction algorithm. The results show that the reconstruction of 35° azimuth sectors feature mean square errors of less than 1 m²/s² and absolute errors of less than 2 m/s in 99% and 98%, respectively, of all cases. The runtime is about 0.1 minutes on average with commercial off-the-shelve CPU hardware. The reconstructed wind field of radial velocities can be used either to fill in sectors where the lidar is blocked e.g. by an obstacle or to extend the maximum operational range by

measuring only a few lines-of-sight with increased pulse accumulation time. An example of a range extension PPI provided here demonstrates that the range can be extended to 25 km while maintaining the total recording time of 30 s as for the reference PPI scan featuring only a maximum range of approximately 12 km.

## 1.1 Introduction

Physics-Informed neuronal networks are the subject of a field that develops rapidly since the work of Raissi (Raissi et al. 2017)

who introduced a PINN data driven approach in 2017. Many acronyms like physics-informed, physics-guided in combination with neuronal networks (NN), deep learning (DL) or machine learning (ML) exist (Ni et al. 2022), but all of them relate to a machine learning approach using a neuronal network to solve a differential equation describing a physical phenomenon. Complex physical phenomena like convection, heat transfer or shock waves can be described by partial differential equations that are difficult to solve. Neuronal networks represent a new capability to approximate a possible solution of these equations

by training a neuronal network.

3D scanning pulsed Doppler lidars are more and more widely deployed at airports to detect dangerous wind phenomena. A large maximum operational range and a fast update rate, i.e. scanning speed, are required to detect wind hazards as early as possible to ensure safe airport operations. At airports, scanning lidars are almost always blocked in some directions due to tall buildings or other safety-related precautions. These areas need to be blanked out which reduces the situational awareness.

Moreover, these data gaps can have a negative impact on meteorological data products that are e.g. based on tracking algorithms.



A large maximum operational range of a coherent pulsed Doppler lidar can be conventionally achieved by increasing the pulse energy or the pulse accumulation time. The former is limited on most lidars based on fiber lasers due to non-linear effects within the amplifier. However, increasing the pulse accumulation time is tantamount to decreasing the update rate of the scan which often is not an option.

In this work, we use a physics-informed neural network to 1) boost the maximum range by using only a limited set of lines of sight with long accumulation time, thus leaving the overall update rate of the scan unaltered and 2) fill in data gaps in blanked sectors.

In the literature, according to the authors' knowledge, only one author Zhang (Zhang & Zhao, 2021), is known who reported

the application of a PINN algorithm to wind lidar measurements. The goal there was to reconstruct a wind field incident on a wind turbine using a non-scanning short-range lidar (with a maximum operational range of 220 m). A wind field was reconstructed with the PINN algorithm using the 2D Navier-Stokes equation in Cartesian coordinates. The results in Zhang's contributions were compared to a numerical simulation. In contrast, a 3D long range scanning pulsed Doppler lidar performing high resolution wind PPI measurements is used in this work. The lidar provides the data used as boundary conditions for the

PINN and it is also used to verify the results of the PINN calculations. The Doppler lidar measures radial velocities with an accuracy of less than 0.5 m/s. Based on data from an almost one-year measurement campaign, the accuracy of the radial wind component reconstructed using PINN is analyzed. In addition, a dedicated scan was set up to extend the range of a full PPI by applying the PINN algorithm.

## 1.2 Physics-informed neuronal networks and the 3D scanning lidar method

The goal of this paper is to describe the coupling between the PINN algorithm with 3D scanning Doppler lidar data to reconstruct empty sections and to extent the maximum operational range. For wind monitoring at an airport, a Doppler lidar usually performs multiple plan position indicator scans (PPI) under different elevation angles by measuring the radial velocity per line of sight (LOS). To reconstruct an empty PPI-sector, only the left and right adjacent lidar LOS measurements serve as an input to the PINN algorithm. The architecture of the physics-informed neuronal network coupled with 3D scanning lidar

data consists of three main blocks as shown in Figure 1. Block 1 shows the neuronal network. For simplicity, only three hidden layers, each of which accommodates seven neurons, are depicted. The real network consists of five hidden layers, each of which possesses fifteen neurons. As the activation function, the rectified linear unit (ReLU) has been chosen. The whole NN configuration is specified in Table 1. The spatial and temporal domain data range (r), azimuth angle (θ) and time (t) serve as inputs for the neuronal network. The output data set consists of the pressure p, the transversal and radial velocity component

$V_\theta$ and $V_r$ suggested as a best-fit solution of the NS equation, given the data.





**Figure 1: PINN set up; block 1) shows the neuronal network, block 2) the physics-informed part, block 3) the scanning lidar data.**

| | |
|---|---|
| **Number of Inputs** | 3 |
| **Number of Hidden layers** | 5 |
| **Number of neurons per Hidden layers** | 15 |
| **Number of Outputs** | 3 |
| **Activation Function** | ReLU |
| **Algorithm/Optimizer** | L-BFGS |
| **Stop criteria/Tolerance Threshold** | 1,00E-06 |
| **Stop criteria/Max. Iteration** | 1500 |

**Table 1: Neuronal network settings**

The NS equation is integrated into the loss function (block 2) together with the adjacent left and right LOS lidar measurement (block3). The non-dimensionalized incompressible NS equation in a polar coordinates system are (Zhu, 2005).


$$\frac{\partial V_r}{\partial t} + \frac{dV_r}{\partial r}V_r + \frac{V_\theta}{r}\frac{\partial V_r}{\partial \theta} + \frac{V_\theta^2}{r} = -\frac{\partial p}{\partial r} + \frac{1}{R_e}\left(\frac{\partial}{\partial r}\left(\frac{1}{r}\frac{\partial(rV_r)}{\partial r}\right) + \frac{1}{r^2}\frac{\partial^2 V_r}{\partial \theta^2} - \frac{2}{r^2}\frac{\partial V_\theta}{\partial \theta}\right) \qquad (1)$$


$$\frac{\partial V_\theta}{\partial t} + \frac{dV_\theta}{\partial r}V_r + \frac{V_\theta}{r}\frac{\partial V_\theta}{\partial \theta} + \frac{V_r V_\theta}{r} = -\frac{1}{r}\frac{\partial p}{\partial \theta} + \frac{1}{R_e}\left(\frac{\partial}{\partial r}\left(\frac{1}{r}\frac{\partial(rV_\theta)}{\partial r}\right) + \frac{1}{r^2}\frac{\partial^2 V_\theta}{\partial \theta^2} + \frac{2}{r^2}\frac{\partial V_r}{\partial \theta}\right) \qquad (2)$$

Where $R_e$ is the Reynolds number, $V_r$, $V_\theta$, denotes the radial und tangential velocity, respectively. For the calculation, all variables will be non-dimensionalized through selection of appropriate scales for the characteristic length L (200 m) and

velocity U (10 m/s) to derive $r_i = \frac{r}{L}$, $V_i = \frac{V}{U}$, $t_i = \frac{t}{\frac{L}{U}}$ and $R_e = \frac{V_i r_i}{v}$ with v is the kinematic viscosity of air (1.5 $10^{-5}$ m$^2$/s ). The calculation of the gradients, needed to solve the NS, were determined using automatic differentiation. To minimize the loss function,

$$\text{Loss Function} = \text{MSE}_{NS} + \text{MSE}_{Lidar} \qquad (3)$$

the mean squared error (MSE) composed of the lidar data and the residual of the NS equation are calculated:


$$\text{MSE}_{NS} = \frac{1}{N_S}\sum_{i=1}^{N_S}\left|f_{NS}(\theta_i^S, r_i^S, t_i^S)\right|^2 \qquad (4)$$

$$\text{MSE}_{Lidar} = \frac{1}{N_{LL}}\sum_{i=1}^{N_{LL}}\left|(V_i^{LL}(\theta_i^{LL}, r_i^{LL}, t_i^{LL}) - V_i\right|^2 + \frac{1}{N_{LR}}\sum_{i=1}^{N_{LR}}\left|(V_i^{LR}(\theta_i^{LR}, r_i^{LR}, t_i^{LR}) - V_i\right|^2 \qquad (5)$$

Where $\theta_i^S$, $r_i^S$, $t_i^S$, are the spatial and temporal data characterizing the PPI sector, to be reconstructed by the PINN-algorithm.

$\theta_i^{LL}$, $r_i^{LL}$, $t_i^{LL}$ are the spatial and temporal data corresponding to the left and $\theta_i^{LR}$, $r_i^{LR}$, $t_i^{LR}$ for the right lidar beam. $V_i^{LR}$, $V_i^{LL}$, are the radial velocities corresponding to the left and right lidar beams where $V_i$ correspond to the radial velocities of the reconstructed domain, i.e. the proposed solution of the NS equation.

$N_{NS}$ is the number of points or rangegates inside the reconstructed domain and, $N_{LL}$, $N_{LR}$ of the lidar beams (left and right). $f_{NS}$ is the residual function of the NS equation. The learnable parameters (weights and biases) within the NN need to be trained by

enforcing that for the inputs ($\theta_i^S$, $r_i^S$, $t_i^S$), the output of the network $V_i$ fulfills the NN equation and the boundary conditions. To minimize the loss, the low-memory Broyden-Fletcher-Goldfarb-Shanno (L-BFGS) optimization algorithm (Liu & Nocedal, 1989) is used. In contrast to non-physical constrained neuronal networks, PINN has to be re-trained by every new input of lidar data. Therefore, in order to minimize the loss function recurrently, the training process must be carried out very quickly for real-time applications. To control the training run-time process, two parameters are specified as truncation criteria: The

optimization algorithm to minimize the loss function stops, if a tolerance of 10⁻⁶ or a number of 1500 iterations is reached.





This leads to a compromise between computation time and accuracy. For wind field reconstruction applications at airports, the PINN algorithm has to work very fast in order to not significantly increase the lidar data update rate.

### 1.3 Radial velocity PPI section reconstruction results

We make use of data recorded during an almost one-year test bed campaign at Frankfurt airport in 2021, where a 3D scanning
lidar performed high resolution wind measurements. The Doppler lidar was positioned on the roof of a parking garage approximately 760 m away from the center runway. We focused our investigation on 360° PPI's taken with an azimuthal resolution of 2.5 degrees. The elevation angle was 1.5 degrees and the range resolution was 150 m. To reconstruct a wind field, a section of 35° (from 70° to 105°) was selected which is quite large for lidar blanked sectors at airports. Figure2 shows two examples of reconstructed PPI sectors using PINN (lower graphs) and, for comparison, the corresponding original
measurement (upper graphs).

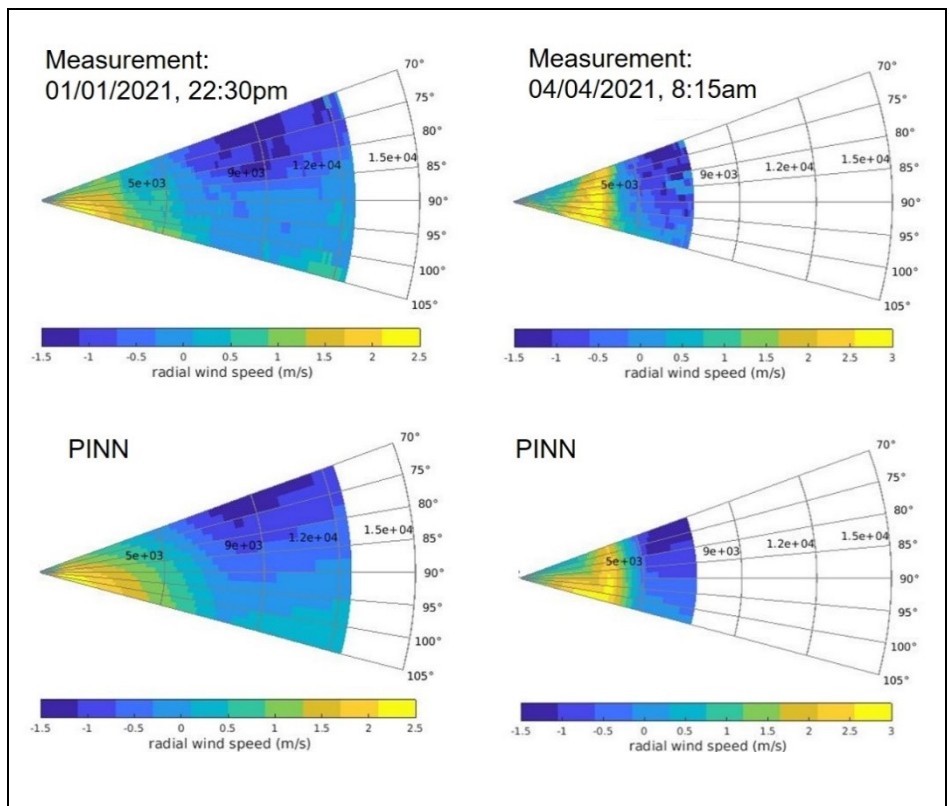

**Figure 2: 35° Sector reconstruction examples, above measurements, below PINN, left: long range, right: smaller range.**

The lidar data maximum range varied due to the prevailing atmospheric conditions. Figure 3 shows a histogram of the maximum range distribution (left) of the sector reconstructed by the PINN algorithm. In most of the cases, the maximum range achieved is between 8 km and 14 km, which means that the distance between left and the right boundary lidar beams increases
significantly as the range becomes larger. Consequently, the PINN algorithm needs to reconstruct greater areas, what is



obviously more challenging. The average computation time is about 0.1 minutes (Figure 3 (right)), which is fast enough to integrate it into a lidar scheduler for real time processing. The analysis was carried out using an ordinary desktop computer equipped with an Intel Xeon 3,8 GHz processor (computation only on one core).

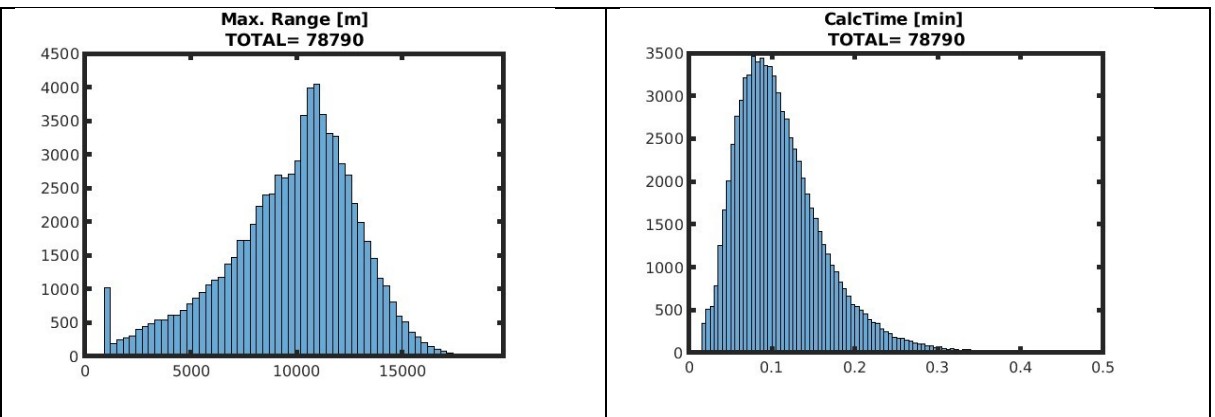

**Figure 3: Left: Histogram (nbin=50) of maximal ranges of the reconstructed sector right: Histogram (nbin=150) of the PINN**
**Algorithm run-time**

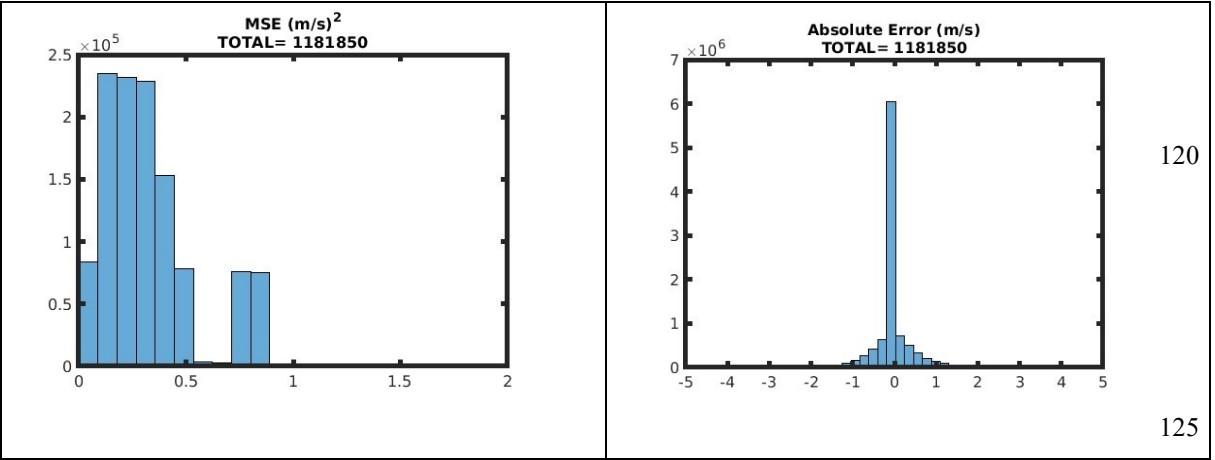

**Figure 4: Left: Histogram (nbin=500) of the mean square error (MSE), right: Histogram (nbin=500) of the absolute error**

The 35° sector consists of 14 LOS's. The reconstructed data have been evaluated by mean square and absolute error of their differences to the actual lidar measurements. In total, more than 1.1 million LOS's were analysed as indicated by the histogram plots in Figure 4. Considering a range gate size of 150 m and the maximum range distribution as shown in Figure 3 (left), more

than five million range gates were compared in total. The result shows that the PINN algorithm is able to reconstruct the radial wind speed in 99 % of all cases with an accuracy of less than $1 m^2/s^2$ for the mean square error and less than 2 m/s for the absolute error in more than 98 % of all cases.



## 1.4 Doppler lidar range extension method

Nowadays, many Air traffic management research initiatives are attempting to increase airport capacity by applying new
concepts for arrivals and departures to optimize wake turbulence separations (EUROCONTROL, 2019), (EUROCONTROL,
2020). For the arrival concepts, for example, wind information for the full glide path is desirable (EUROCONTROL, 2020).
The range of a coherent pulse Doppler lidar can be extended by increasing the pulse accumulation time. However, this makes
the measurement of a full PPI very slow. Nevertheless, by measuring only at sparse azimuth angles, the accumulation time per
LOS can be increased while the total PPI measurement time remains the same. PINN can be used to fill the empty sectors in
between to receive a complete PPI with extended range. Figure 5 left shows a dedicated scan (intermitted scan) sampling only
at 10 locations with a scanning speed of 0.33°/s. Figure 5 right shows a reference PPI recorded with 1 degree azimuthal
resolution and a scanning speed of 12°/s. Both scans take approximately 30 s in total. The calculation time of the PINN is
neglected in this consideration, which can be achieved through various means (see discussion section). The center of Figure 5
shows the completely filled PPI using PINN. The result demonstrates that the range can be extended from approximately 12
km to 25 km. In this use case, due to the long range, a very low elevation had to be chosen in order to not be limited by clouds
or the atmospheric boundary layer. The ten azimuthal beams were chosen so that they were not shadowed by tall buildings or
woods but still provide a good representation of the PPI.

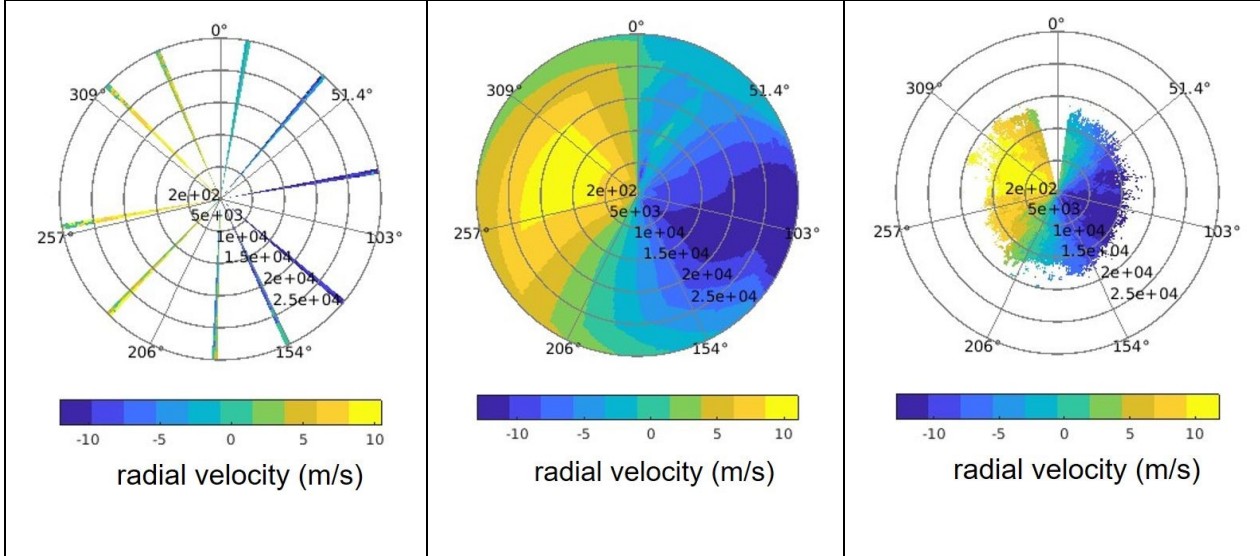

**Figure 5: Left: Range extension scan, middle: PINN reconstructed full 360° PPI, right: reference scan.**

Another advantage of the method is that the lag-angle effect is minimized. The lag-angle effect is caused by the mismatch
between the receiving aperture and the backscattered signal that occurs when the scanner rotates rapidly. Fast rotary movement
of the scanner is necessary for the airport application in order to achieve the required high update rate of a PPI. The faster the
scanner rotates, the greater the mismatch. At a typical angular speed of 12 °/s, the size of the effect amounts to a loss of about
3 dB in signal to noise ratio at 10 km range, corresponding to ca. 2 km less maximum operational range. With the range





extension scan, fast scanning is only required between the measurements of the sparse beams which has no influence on the

155 measurement itself.

### 1.5 Dependence of reconstruction quality on number of lidar beams: Assessment of a gust front use case

The range extension method used in an operational environment bears the risk that dangerous wind phenomena are missed in the case of too large empty sectors. This depends on the typical scale of such phenomena in the local micro-climate of the airport. It is therefore advisable to characterize such events thoroughly and choose the sparsity of the range extension scan

160 employed accordingly. The possibility that a hazardous phenomenon can neither be detected by the sparse LOS measurement nor reconstructed by the PINN algorithm has to be ruled out for safety reasons. In order to miss a phenomenon completely, the inhomogeneity caused by forcings outside the boundary of the domain on which the NS equations are solved needs to fit completely between two LOS's. Due to continuity constraints of the flow field, such phenomena cannot be arbitrarily small and severe at the same time which means that it should be possible to locally define an adequate scan.

165 Here we show as an example a gust front use case scenario, which occurred on July 10th 2021. For this case, the complete PPI has been reconstructed using different numbers of LOS beams and subsequently compared with the measurement. At 20:30, an inflowing gust from the southwest was detected in the measurement PPI, which then passed over the airport area and caused a wind shear event at 20:45 hrs. For these two timestamps, the PPI has been reconstructed with 8 (with 45° empty sectors), 12 (with 30° empty sectors) and 16 (with 22.5° empty sector) LOS beams, starting from zero.

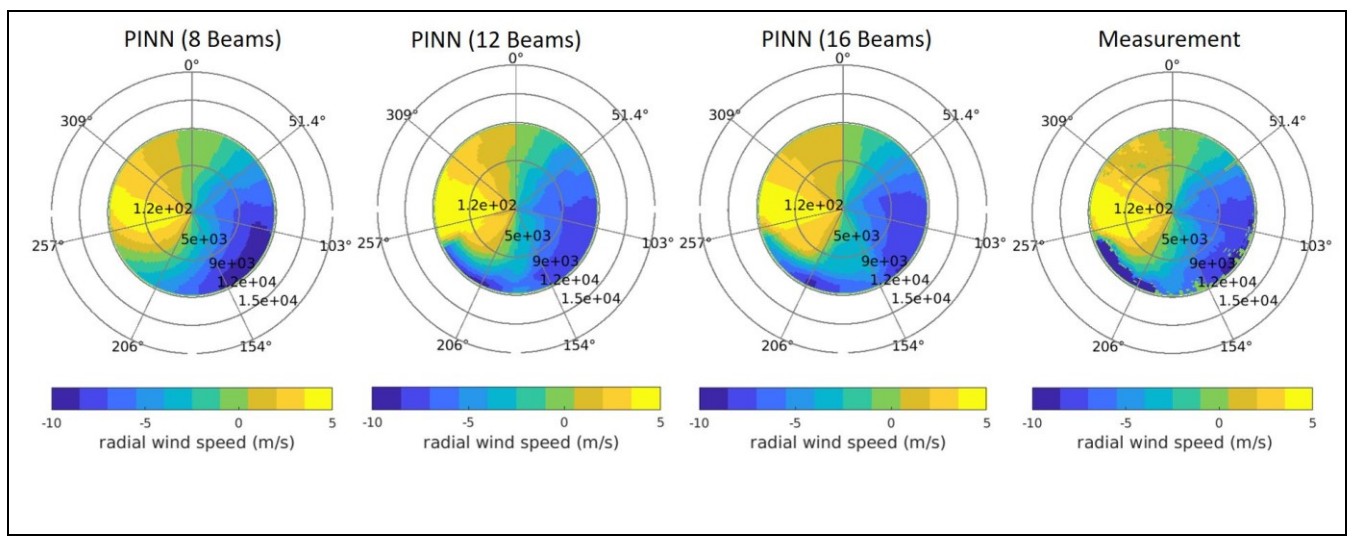

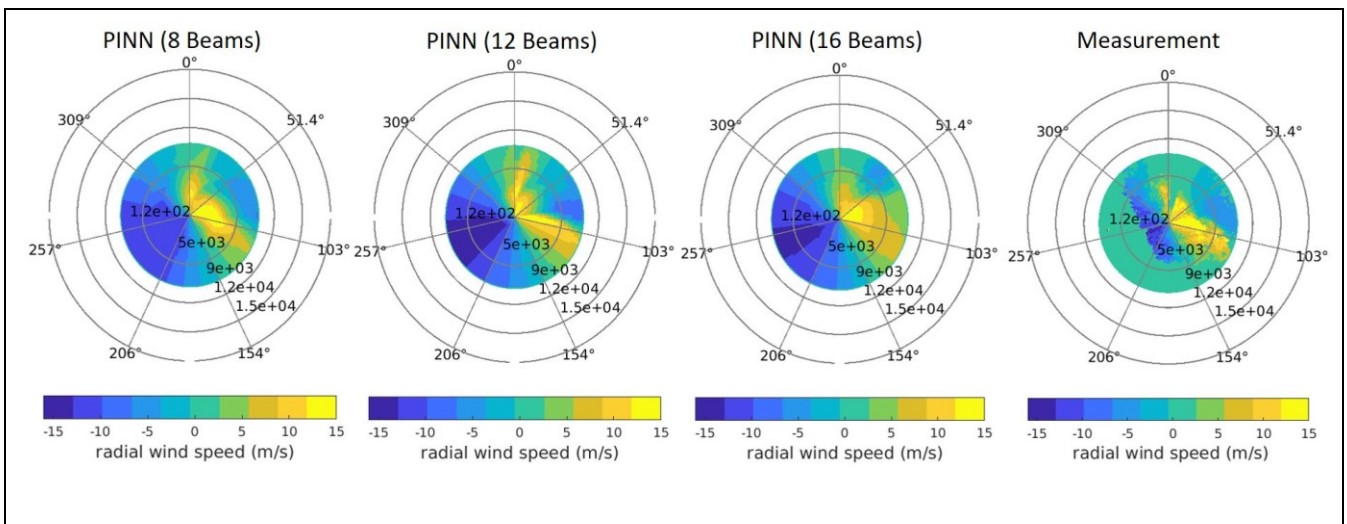

**Figure 6: From left to right: PINN-reconstructed PPI using 8 Beams, 12 Beams and 16 Beams, lidar Measurements; Above for the gust front use case @ 20:30hrs, below: Wind shear use case @ 20:45hrs.**

Figure 6 shows that for both use cases (gust front and wind shear), using 8 to 10 LOS beams are sufficient to reproduce the PPI and resolving the phenomenon well enough for the detection. Of course, these are only two examples and cannot be generalized to all dangerous wind phenomena. However, as explained, it is also a fact that the wind hazards for aircraft must have a certain spatial extent correlated with their strength, which can be captured with a corresponding amount of LOS beams. Furthermore, the lidar continuously scans PPIs at the airport, so if the range extension method misses a phenomenon, the other scans will catch it, but just at that range that pertains to the normal maximum operational range of the lidar. Therefore, the intermittent scan may be regarded as a kind of early detection scan, allowing the detection of dangerous wind situations at longer ranges. This way, nothing is really lost, but only additional information gained.

As mentioned above, the calculation time of the PINN algorithm has not been considered in the range extension use case. A real time application of this method can be in principle achieved through various measures. Using more computational power is the straight forward approach, while the calculation on CUDA cores is a different option. For the latter, though, a high degree of parallelization of the algorithm will be necessary.

## 1.6 Discussion

Artificial intelligence methods as tools for enhancing and extending measurement data abound today. Physics induced neuronal networks are special in that they do not depend on data mining techniques on long historical data sets, but on rigorous physical laws. Therefore, problems like incorrectly interpreted correlations due to insufficient training data are not an issue here. Instead of having a trained network for all future inputs, the network is optimized each time from scratch to find an adequate solution that fits the NS equation based on the updated boundary conditions. This way of solving a partial differential equation also





differs significantly from the computational fluid dynamics (CFD) approach, where the NS equations are solved numerically based on some assumptions and simplifications.

With PINN, one relies on the fact, that a flow field is completely determined by the initial and boundary conditions, if transient features inside the domain that are not reflected on the boundary are absent, i.e. the flow is stationary. If such cases can be excluded up to some required strength, data reconstructed in domains not covered by measurements can be very much relied

on, as the statistics on artificially blanked sectors of scanning Doppler lidar PPI scans presented here in 1.3 show. By the same token, the application of the PINN method presented in this contribution to the range extension of low elevation Doppler lidar PPI scans can be considered reliable under the assumption of quasi stationarity. It does not present an extrapolation which is, of course, always at one's own risk, but is basically the same method as the reconstruction of the blanked sector combined with a different scan strategy for the lidar. It may also be classified as compressed sensing for that matter, since it relies on

reducing the density of measurements according to the actual information content contained in the data. As explained in 1.5, this information content dynamically changes in a real atmospheric flow field and different spatio-temporal sampling of data may be necessary to obtain the full set of required features present. In order to use the range extension operationally, one may envision dynamical scans that are adapted according to the weather situation, but one may also devise a statical approach due to the fact that the severity of wind phenomena and their spatio-temporal extent are correlated. Additionally, it should be noted

that an operational lidar scans multiple PPI in a loop, so if a hazardous phenomenon is missed in the range extension method, the other scans would detect it, but at normal operational range.

## 1.7 Summary and conclusion

In this publication, the coupling of a physics-informed neuronal network with a 3D scanning pulsed Doppler lidar has been presented. The Navier Stokes equation in polar coordinates was implemented to reconstruct radial velocity sectors of PPI

scans. To this end, only the lines of sight adjacent to a particular angular sector were used as input. Almost one year of data from a measurement campaign at Frankfurt airport in 2021 was used to assess the performance of the PINN algorithm developed to reconstruct the radial wind field. Wind field sectors of 35° azimuthal width with a maximum range of up to approximately 15 km have been reconstructed and compared with the measurement of a 3D pulsed scanning Doppler lidar. More than five million range gates have been considered for the analysis, showing that the reconstructed wind field values

deviate with an MSE of less than 1 $m^2/s^2$ compared to the measured values. Using the same method with a special scan pattern of the lidar featuring a sparse array of lines of sights with increased pulse accumulation times, the maximum operational range of the full PPI could be significantly increased to about 25 km, leaving the overall scanning time unaltered. The fact that the physics of the flow field governed by the NS equation is uniquely determined given by the boundary and initial conditions, the analysis of a gust front use case leaves us confident that the range extension method can be operationally used to enhance

airport wind field monitoring. However, this scan should be combined with normal PPI's in the lidar scheduler for back up reasons. In addition to the performance gain, the PINN algorithm has the advantage that it does not require large sets of training data, as conventional machine learning algorithms do. This means that the algorithm can be applied immediately without prior



data collection. Issues normally encountered with data driven approaches like the learning of false correlations specific to the training data and not the relevant process are absent here.

Since the algorithm has to be re-trained with each new measurement of the Doppler lidar, the training time plays a critical role for the real-time application. The calculation time was determined to be about 0.1 minutes on average when using ordinary PC hardware.

The quality of the transversal velocity component also provided by the algorithm could not be assessed, since its validation was outside the scope of this work and appropriate reference data were not available. Should this turn out to be of good or

acceptable quality, this additional output would be very useful as it would efficiently solve the "cyclops dilemma" of the Doppler lidar technique, which measures only the radial component of the flow vector. It should be noted that this was the main aim of Zhang (Zhang & Zhao, 2021), to implement the PINN method in the first place.

**Competing interest**

The contact author has declared that none of the authors has any competing interests.

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
