# Peer review of "Coupling physics-informed neuronal networks with 3D scanning pulsed Doppler lidar"

_Atmospheric Measurement Techniques, 2023_

## Referee Comment (RC2)

**Review of: "Coupling physics-informed neuronal networks with 3D scanning pulsed Doppler lidar"**

This paper used a Physics-Informed Neural Network (PINN) to attempt to reconstruct a wind field using sparse measurements from a scanning doppler lidar. The authors present a method for filling in missing data using the polar Navier–Stokes equations as boundary conditions. The method is evaluated for two cases: 1) reconstruction of missing lidar data and 2) extension of lidar range using integration and reconstruction over sparse regions.

I am not convinced that this method accurately estimates the winds, and I am unsatisfied with the presentation of the statistics that assess the accuracy and precision. The authors need to take more care in how they evaluate the reconstructed winds vs. measured winds, how they assess separately the uncertainty, variability, and bias, as well as how sampling and range contribute to the final intercomparison. I suggest that a few of these major issues be addressed before publication.

**Major Comments:**
**1a)** Much more care needs to be taken when comparing the "true" measured winds to the reconstructed winds. The generic term "errors" is encompassing too much and makes reading this paper frustrating. Please be sure to use consistent and clearly defined terms to separate between:
"Measurement Uncertainty" (Poisson, instrument, and retrieval error),
"Residuals" (difference as a function of range and azimuth between the measurement and reconstruction)
"Bias" (systematic tendency in the residuals between measurement and reconstruction),
"Accuracy" (comments on the magnitude of the Bias or Residuals)
"Precision" (comments on the magnitude of the measurement uncertainty)
"Significance" (assessment of the bias with respect to the precision of the measurement)

**1b)** Use RMSE instead of MSE, as it will put your results in the same units as the measurement (m/s). It is more intuitively useful for the reader to know the bias rather than the square of the bias, which has no physical meaning. Figure 4 is extremely misleading! Must be range dependent or quoted at a few characteristic ranges.

**1c)** When presenting the final results (line 12 and conclusion) the median plus/minus the standard error should be given, not the 98th and 99th percentile. Currently, it presents a skewed picture to the reader and makes it hard to assess whether the comparison is actually any good.

**1d)** Line 45: You have reported a scalar 'accuracy' of less than 0.5 m/s. This tells me nothing as a reader. Given the higher data density closer to the lidar, a single value for bias will be range-weighted. Accuracy needs to be assessed as a function of range and azimuth.

**2)** Lines 73-75:  I'm uncertain why these particular values for characteristic length and velocity are chosen.  Please consider adding more text to explain your choice.

**3)** I'm unclear about your Loss Function in Eq. 3.  Why have you chosen to minimize the MSE rather than a more standard technique of Least Squares.  As the Loss function is written, it appears to always minimize the sum of two positive numbers based only on the boundary condition lidar profiles and the retrieval.  This doesn't seem like it will work very well at larger lidar ranges where there is a significant gap between the data.  It could lead to results where the data contributes very little.

**4)** Figure 5 looks very strange.  The reconstructed wind field does not match the measurements on the left and has an "unphysical" polar coordinate rectangle shape for the yellow wind gust.  This tells me that there is insufficient data to influence the retrieval grid.  A second quality check that could be done is looking at the left panel at the beams near 257 and 309 degrees.  At the outer ring, the wind speeds are low (teal-green, 0-5 m/s).  In the PINN the outer ring on this segment starts at yellow and goes to green.  Why does the PINN not match the boundary point values provided by the data?

**5)** Figure 6 (lower panel for 20h45) makes me very skeptical about what you are doing!!! The PINN reconstruction for 12 and 16 beams has a very large (> -15 m/s) radial wind speed around the 257 degrees mark that doesn't exist in the data!!!!

**6)** Multiple times while reading the paper, I wanted to ask how does this PINN method compare to simply interpolating between adjacent lidar profiles? Does PINN give a meaningfully better result than simple interpolation?

**Minor Comments:**
L13-14: Please use seconds to be consistent with later text (L12 and L111)

L26:    Please cite a paper regarding a typical lidar to which this method could be applied.

L28:    Please cite a paper and tell the reader what the safety requirements are.  Do airports require wind information every second?, 10 seconds?, every minute?  And at what resolution?  This helps the reader evaluate the required resolution of the lidar scans.  For example, the system is over-engineered if you produce a complete scan every 1s at 1.5 m range gating, but the airports only require 60 s at 100 m.  Then it's better to drop the resolution and speed of your system and reduce the Poisson error.  Same comment on L97 and L174:  How fast does the system need to work to satisfy user requirements?

L51:    "extent" → "extend"

All figures were a bit "granular" on the PDF.  Please save figures at a higher resolution (300 dpi) and include larger fonts on axis labels

L112:   What is a "lidar scheduler"?

Figure 3 (left) lacks relevance or motivation. Does this tell me about range limitations in the hardware for signal acquisition, local geographical gaps, or about the algorithm?

Figure 3 (right) is OK, but why does it need a figure? It's good enough to describe as text

L142-143: This sentence is a bit unclear.  Please consider rephrasing.

Figures 5 and 6 need clearer range information included either in the figure or caption.

L160-161:  You could try to mitigate this by offsetting the pointing direction on each rotation (ie. non-harmonic of π azimuthal step size). Similar to a satellite with a precessing orbit.

Figure 1 is not useful.  You have tried to fit too much, and it is difficult to interpret.  I had to spend time to understand the figure, rather than the figure aiding my understanding of the text. Please consider revising it to something simpler.  I only need to know: Measurement ($\Theta$, r, t) → "black box" → Retrieved ($v_r$, $v_\Theta$, P) using Navier-Stokes (NS) and a loss function.  The NS equations are better presented in Eq. 1 and Eq.2 than in the figure.

Table 1 doesn't add to the paper.  The information is in the text.  Please move to an Appendix or remove

L175: In essence, you have a reliable data product at short-range, then you use some sparse scans to attempt to extrapolate further.  There is nothing wrong with this, but perhaps adding a red line in the polar plot to mark a "reliability" limit might be helpful.  Calculating and showing your median bias and standard error will allow you to assess over what range your reconstruction is significantly equal to the measurement.

L183:  Is this future work?  You can already test if parallelization gives a large performance boost on your laptop. I generally test/develop lidar code locally on 4 to 6 cores using my work laptop, then allow many more cores on the server for batch processing years of data. It is very straightforward to try.

L195: Section 1.3?

L197-200:  A bit unclear.  Please consider rephrasing. Compressed sensing wasn't mentioned in the introduction, so it doesn't really fit in the discussion.

L200: Section 1.5?

L230:  I have no idea what the "Cyclops dilemma" is.  Either explain it in the introduction or remove it.

L231-232:  The last line of your conclusion is rather weak.  Comparison with (Zhang and Zhao, 2021) should be in the discussion section.

---

## Author Comment (AC2)

Review of: "Coupling physics-informed neuronal networks with 3D scanning pulsed

Doppler lidar"

*Dear reviewer, thank you for taking the time to provide your input. Please find below our detailed responses to your comments.*

This paper used a Physics-Informed Neural Network (PINN) to attempt to reconstruct a wind field using sparse measurements from a scanning doppler lidar. The authors present a method for filling in missing data using the polar Navier–Stokes equations as boundary conditions.

*This is a misunderstanding. We do not use the Navier-Stokes (NS) equation as Boundary Conditions (BC). Instead, we use the Lidar data as BC for the NS equation. This is explained in L44. "The lidar provides the data used as boundary conditions for the PINN and it is also used to verify the results of the PINN calculations."*

The method is evaluated for two cases: 1) reconstruction of missing lidar data and 2) extension of lidar range using integration and reconstruction over sparse regions.

I am not convinced that this method accurately estimates the winds, and I am unsatisfied with the presentation of the statistics that assess the accuracy and precision. The authors need to take more care in how they evaluate the reconstructed winds vs. measured winds, how they assess separately the uncertainty, variability, and bias, as well as how sampling and range contribute to the final intercomparison. I suggest that a few of these major issues be addressed before publication.

*We assume that the general critics provided above are detailed in the next comments. If this is not the case we kindly ask for more specific inputs.*

Major Comments:

1a) Much more care needs to be taken when comparing the "true" measured winds to the reconstructed winds. The generic term "errors" is encompassing too much and makes reading this paper frustrating. Please be sure to use consistent and clearly defined terms to separate between:

"Measurement Uncertainty" (Poisson, instrument, and retrieval error),

"Precision" (comments on the magnitude of the measurement uncertainty)

"Significance" (assessment of the bias with respect to the precision of the measurement)

"Bias" (systematic tendency in the residuals between measurement and reconstruction),

 "Accuracy" (comments on the magnitude of the Bias or Residuals)

*We did not use "error" as generic term. Instead we have used the terms "mean square error" and "absolute error". These terms are precisely defined in textbooks and are commonly used if two data sets are compared (in this case measured vs. reconstructed data).*

"Measurement uncertainty" and "precision" can only be applied to measurements, which are not the topic of this publication. We will make more explicit the specification of the lidar used, but will not discuss it here. However, we will provide the reader with some insight on the influence of variability in the data and the effect of "outliers" for which to suppress, care needs to be taken indeed. Hence, this will certainly add to the value of the paper.

"BIAS" is specified only once as a parameter for the neural network (Weights and Biases - AI Wiki (paperspace.com)). Weights and bias are both learnable parameters inside the network. During the training of the neuronal network, both parameters are adjusted toward the desired values and the correct output.

 From our perspective, "Significance" is a statistical term which is not very useful to describe the result of this work. It can be only derived in relation to a Null-Hypothesis for which a probability p is to be assigned to produce the data. The physical process here is known to be the flow field of wind as measured by a Doppler Lidar. To assume, for example, that instead a random process of i.i.d. values has produced the data is far fetched but, more importantly, irrelevant for the work at hand. We only need to show the reconstructive power of the algorithm of the data given which is completely described by the Model vs. Measurement statistics. Therefore, we will elaborate more on the latter, especially with regard to its dependency on range and azimuth, as this has been also requested by Mr. Zhang.

However, the term "accuracy" is erroneously used in the sentence  "The result shows that the PINN algorithm is able to reconstruct the radial wind speed in 99 % of all cases with an accuracy of less than $1m^2/s^2$ for the mean square error and less than 2 m/s for the absolute error in more than 98 % of all cases. "  and in 1d) These sentences will be modified.

 "Residuals" (difference as a function of range and azimuth between the measurement and

reconstruction)

In the text the term "residuals" is used to denote the numerical deviations in the solution of the governing equation (NS), referring to " Residual of the approximation of a function"  (Residual (numerical analysis) - Wikipedia).This usage is very common and accepted in scientific literature(see e.g. reference Zhang et. al, or https://doi.org/10.1063/5.0095270, Therefore, we will not speak of residuals to signify the Model-Measurement difference, but suggest to use the term "observation – reconstruction statistics" following the term "Observation minus background" statistics common in the numerical weather prediction context.

1b) Use RMSE instead of MSE, as it will put your results in the same units as the measurement (m/s). It is more intuitively useful for the reader to know the bias rather than the square of the bias, which has no physical meaning. Figure 4 is extremely misleading! Must be range dependent or quoted at a few characteristic ranges.
MSE and RMSE are both recognized quantities in error statistics. We use the mean squared error because it is a common way to measure the prediction accuracy of a model vs. ground truth (measured) data. MSE is an error metric where the closer the value is to zero, the more accurate the model performs. Nevertheless, to make it easier to understand, we also present the absolute error in the manuscript.

1c) When presenting the final results (line 12 and conclusion) the median plus/minus the

standard error should be given, not the 98th and 99th percentile. Currently, it presents a skewed picture to the reader and makes it hard to assess whether the comparison is actually any good.

We have used the percentiles in order to show the quality of the reconstruction with respect to outliers. We believe they are much better suited for this purpose than median and standard deviation, since the supremum is of more interest here. However, we will add a comment explaining our choice in the text especially with regard to safety critical applications where one wants to know what is the worst case one has to deal with.

1d) Line 45: You have reported a scalar 'accuracy' of less than 0.5 m/s. This tells me nothing as a reader. Given the higher data density closer to the lidar, a single value for bias will be range-weighted. Accuracy needs to be assessed as a function of range and azimuth.

"Accuracy less than 0.5" is the manufacturer's specification of the Doppler lidar. We take the measurement as ground truth, but all measurements have an error, we wanted to make this clear. We will rephrase it as "The manufacturer specifies a maximum error of 0.5 m/s for the Doppler lidar which provided the measurements used in this publication "and insert a reference.

2) Lines 73-75: I'm uncertain why these particular values for characteristic length and velocity are chosen. Please consider adding more text to explain your choice.

We have estimated our characteristic length L and velocity according to the assumptions of meteorological scale analysis. We choose the L according to the Rosby number which has to be significantly larger than one for non geostrophical wind (boundary layer phenomena) for which the Coriolis forces can be neglected. Additionally, we did some trial and error analysis over several lengths in order to optimize our choice based on the results. The best results in terms of accuracy have been achieved for L=200m. We will consider this in an additional sentence.

3) I'm unclear about your Loss Function in Eq. 3. Why have you chosen to minimize the MSE rather than a more standard technique of Least Squares. As the Loss function is written, it appears to always minimize the sum of two positive numbers based only on the boundary condition lidar profiles and the retrieval. This doesn't seem like it will work very well at larger lidar ranges where there is a significant gap between the data. It could lead to results where the data contributes very little.

The MSE is the common metric for the Loss Function in testing maching learning methods, see e.g. Ref. P. Ni et. al. in our references. We will add a comment on that.

As Fig. 4 and others we will add with regard to the discussion above show that the method works objectively. We take this to be more relevant than gut feeling. Apart from that, the gap growing with range is inherent to the problem's polar symmetry which is respected in the NS equation in polar coordinates. By the nature of this being a second order partial differential equation, it's solution is determined by boundary and intial conditions, no matter the size of the domain over which the boundary is defined. What increases with an increasing gap, is the chance of having a phenomenon happening inside of it that is not represented on the boundary and will therefore not appear in the solution. We will make this more clear in the text.

4) Figure 5 looks very strange. The reconstructed wind field does not match the measurements on the left and has an "unphysical" polar coordinate rectangle shape for the yellow wind gust. This tells me that there is insufficient data to influence the retrieval grid. A second quality check that could be done is looking at the left panel at the beams near 257 and 309 degrees. At the outer ring, the wind speeds are low (teal-green, 0-5 m/s). In the PINN the outer ring on this segment starts at yellow and goes to green. Why does the PINN not match the boundary point values provided by the data?

The reason for the mismatches which you have observed is the granularity of the color bar which is quite coarse. We will add a comment on that. In general, the reconstructed PPI captured the reference PPI quite well. However, as you have noticed, there is a mismatch at 257° azimuth. A check has revealed an outlier in the reference data which is probably the explanation for the mismatch. As a matter of fact and already mentioned above, we are aware that the reconstruction method is sensitive to outliers and take this as an opportunity to comment on that. In principle, this a problem that can be solved by lidar data processing and pre-filtering for outliers.

5) Figure 6 (lower panel for 20h45) makes me very skeptical about what you are doing!!! The PINN reconstruction for 12 and 16 beams has a very large (> -15 m/s) radial wind speed around the 257 degrees mark that doesn't exist in the data!!!!

Please consider, the green ring in the Measurement PPI is not zero, it is no data, this is in deed misleading. We will replot the figure clearly distinguishing between data and no-data.

6) Multiple times while reading the paper, I wanted to ask how does this PINN method compare to simply interpolating between adjacent lidar profiles? Does PINN give a meaningfully better result than simple interpolation?

Yes, this could be a new study to compare different Interpolation algorithms. But this is out of scope of this paper. Please consider, the most important point for the user is the comparison to the measurement.

Minor Comments:

L13-14: Please use seconds to be consistent with later text (L12 and L111)

L26: Please cite a paper regarding a typical lidar to which this method could be applied.

L28: Please cite a paper and tell the reader what the safety requirements are. Do airports require wind information every second?, 10 seconds?, every minute? And at what resolution? This helps the reader evaluate the required resolution of the lidar scans. For example, the system is over-engineered if you produce a complete scan every 1s at 1.5 m range gating, but the airports only require 60 s at 100 m. Then it's better to drop the resolution and speed of your system and reduce the Poisson error. Same comment on L97 and L174: How fast does the system need to work to satisfy user requirements?

@ all above minor comments: The method of sector blank reconstruction can be in principle apply to all scanning lidars. We do not want to highlight any manufacturer. However it is possible to design a scanning Doppler lidar that meets the actual safety

requirements like e.g. the EU Directives. The lidar which was used to take the data used in this paper is such a product. In general, no algorithms applied to lidar measurements should not slow down the operational system nor reduce its performance.

L51: "extent" → "extend"

All figures were a bit "granular" on the PDF. Please save figures at a higher resolution (300 dpi)

and include larger fonts on axis labels Your request is understandable, but we believe that the chosen granularity is the best compromise between readability and plot accuracy. For a finer resolution the pictures must be larger.

L112: What is a "lidar scheduler"? A lidar scheduler is the data set which controls the parameters defining the 3D scan such as azimuth and elevation speed and acceleration, PRF, pulse width, elevation step size etc. We will add an explanation.

Figure 3 (left) lacks relevance or motivation. Does this tell me about range limitations in the

hardware for signal acquisition, local geographical gaps, or about the algorithm?

This plot shows the distribution of the measured maximum ranges, which meet the specified error. As I wrote L107: Figure 3 shows a histogram of the maximum range distribution (left) of the sector reconstructed by the PINN algorithm. In most of the cases, the maximum range achieved is between 8 km and 14 km, which means that the distance between left and the right boundary lidar beams increases significantly as the range becomes larger. Consequently, the PINN algorithm needs to reconstruct greater areas, which is obviously more challenging.

Figure 3 (right) is OK, but why does it need a figure? It's good enough to describe as text. The algorithm must be re-trained for each sector, thus the calculation time is not constant. This must be represented. In this case we have decided to show a plot than just drop a value. We believe it is valuable information.

L142-143: This sentence is a bit unclear. Please consider rephrasing.

We will rephrase the sentence.

Figures 5 and 6 need clearer range information included either in the figure or caption.

Fig5 and 6 includes range information, starting from the first range gate up to maximum range. We would need more information as to what is meant by "clearer".

L160-161: You could try to mitigate this by offsetting the pointing direction on each rotation (ie.

non-harmonic of π azimuthal step size). Similar to a satellite with a precessing orbit.

Thank you very much for this hint. We will consider it carefully and possibly test it, if feasible.

Figure 1 is not useful. You have tried to fit too much, and it is difficult to interpret. I had to

spend time to understand the figure, rather than the figure aiding my understanding of the text.

Please consider revising it to something simpler. I only need to know: Measurement (Θ, r, t) →

"black box" → Retrieved (vr, vΘ, P) using Navier-Stokes (NS) and a loss function. The NS

equations are better presented in Eq. 1 and Eq.2 than in the figure.

Unlike Big data ML algorithms, PINN is not a black box. It is much easier using this figure to understand how the algorithm calculates the result. We have used it already in several presentations and received a quite positive response. Therefore, we prefer to keep the figure. However, we will remove the NS equation since it is provided in the text as Eqs (1) and (2)

Table 1 doesn't add to the paper. The information is in the text. Please move to an Appendix or

Remove

Tables need to serve clarity by collating information in a specific order. We believe that table 1 satisfies this requirement.

L175: In essence, you have a reliable data product at short-range, then you use some sparse

scans to attempt to extrapolate further. There is nothing wrong with this, but perhaps adding a

red line in the polar plot to mark a "reliability" limit might be helpful. Calculating and showing

your median bias and standard error will allow you to assess over what range your

reconstruction is significantly equal to the measurement.

Maybe here is a misunderstanding, the "Sparse scan" (fig5 left) is the real measurement, no extrapolation! We reconstruct the empty sectors (no measurement) in between. Fig5 only shows a use case of how to use the algorithm to reconstruct a full PPI with extended range. For the extended range we do not have a reference measurement to quantify the error. This could only be achieved with an additional source (sensor). Please see also Comments and our reply #(1) of the reviewer (Zhang)nr1.

However, if we record data between the spokes (which then have a significantly reduced range due to the accelerated rotational movement) in future, then we will use them in the reconstruction and also mark from when the reconstruction takes place. But as I said, in Fig5 left, we did not record any data between the spokes.

L183: Is this future work? You can already test if parallelization gives a large performance

boost on your laptop. I generally test/develop lidar code locally on 4 to 6 cores using my work

laptop, then allow many more cores on the server for batch processing years of data. It is very

straightforward to try.

In neural network training, parallelization is primarily achieved through model parallelism, where the neural network is distributed across different processors, and data parallelism, where training examples are distributed across different processors and updates to the neural network are computed in parallel. Model parallelism allows training neural networks larger than a single processor can support, but it usually requires adapting the model architecture to the available hardware. I would not consider this to be straightforward. In contrast, data parallelism is model independent and can be applied to any neural network architecture. However, PINN does not use a large training data set. There are publications on this, and it is an option, however we consider computing on CUDA cores to be more efficient. We will add a paragraph explaining the problem of NN-parallelization.

L195: Section 1.3?

Referring to 1.3 is correct here.

L197-200: A bit unclear. Please consider rephrasing. Compressed sensing wasn't mentioned

in the introduction, so it doesn't really fit in the discussion.

L197-200 explain that the range extension method is not an extrapolation. It is only the reconstruction between two adjacent LOS measurements, which was examined in section 1.3. Referring to "Discussion Section for Research Papers, by Rowan Dunton San José State University Writing Center the "compressed Sensing" is mentioned according #5 "Recommend a few areas where further investigation may be crucial."

L200: Section 1.5?

Referring to 1.5 is correct here.

L230: I have no idea what the "Cyclops dilemma" is. Either explain it in the introduction or

remove it.

We will explain the cyclops dilemma. It refers to the fact that single Doppler lidar (and radar) provides radial velocity only.

L231-232: The last line of your conclusion is rather weak. Comparison with (Zhang and Zhao,

2021) should be in the discussion section.

The last section is about the fact that although the PINN algorithm outputs both velocity components, we have only dealt with the radial velocity component because we are limited to it in terms of verification by the measurements of the lidar. This is the mentioned "cyclops dilemma". However, the transverse wind component was verified in the referenced paper. It is not a comparison with Zhang et al., but rather an outlook on future work and thus belongs in the conclusion chapter.